# Effect Mechanism of Extracurricular Tuition and Implications on “Double Reduction” Policy: Extracurricular Tuition Intensity, Psychological Resilience, and Academic Performance

**DOI:** 10.3390/bs13030217

**Published:** 2023-03-02

**Authors:** Chengzhe Fu, Haolin Ou, Tingyang Mo, Liao Liao

**Affiliations:** 1School of Politics and Public Administration, South China Normal University, Guangzhou 510000, China; 2School of Government, Sun Yat-sen University, Guangzhou 510000, China

**Keywords:** extracurricular tuition (ET), “double reduction” policy, psychological resilience (PR), propensity score matching (PSM), intermediate effect

## Abstract

Since its implementation in July 2021, the “double reduction (the reduction of the excessive academic burden on primary and middle school students, and the over-heated off-campus tutoring)” policy (“the Policy”), which aims to foster a sound environment for extracurricular education, has attracted much social attention. At the preliminary stage where data and cases are insufficient, it is of theoretical and practical significance to evaluate extracurricular tuition and its effect mechanism before the Policy was implemented. This study analyzed the said question based on the data from the China Education Panel Survey between 2014 and 2015 (CEPS2014–2015) by virtue of propensity score matching (PSM) and the intermediate effect. The research found that: (1) extracurricular tuition (ET) is positively correlated with academic performance (AP); (2) high and low extracurricular tuition intensities (ETIs) show no difference in improving academic performance, and the group analysis shows that extracurricular tuition is influenced by the “optimal starting zone (OSZ)”; and (3) psychological resilience (PR) is an intermediate variable of extracurricular tuition, which acts as a “conversion valve”. Based on the above findings, the research raised the following suggestions. Firstly, we suggest to emphasize the promotion effect of after-class service on students’ academic performance and switch the functions of extracurricular tuition to after-sale service. Then, pay attention to the reverse effect of high extracurricular tuition intensity on students’ physical and psychological health and their academic performance. In practice, explore the effective “optimal starting zone” of extracurricular tuition for students. Lastly, cultivate students’ psychological resilience by shifting the core of education quality improvement from simple stacking of disciplinary knowledge to psychological health development.

## 1. Introduction

In recent years, as academic pressure has increased, the demand for extracurricular teaching and tutoring is on the rise. Extracurricular tuition (Extracurricular tuition refers to the supplementary education activities beyond the formal school education to improve students’ academic performance. It can be considered as a special form of shadow education, which is private and supplementary, that is, the supplementary teaching provided by private individuals and institutions, which needs to be paid, beyond the formal school education. The “double reduction policy” focuses on the reasonable development of the extracurricular tuition.) has become popular among students and parents [1]. Known as “shadow education” in the academia, extracurricular tuition has expanded its customer base and industry scale. Meanwhile, it results in an overburden for students, anxiety among parents, and a pursuit of profit. Such problems are increasingly worsened, thus seriously deteriorating the edu-ecology and affecting adolescents’ physical and psychological development [2]. For this reason, the 19th Committee of the Central Comprehensively Deepening Reforms Commission deliberated and approved the Opinions (This document, also known as the “double reduction policy”, aims to reduce the students’ heavy homework burden, improve the school’s after-school service level, and comprehensively regulate the off-campus training behavior. At the same time, the Ministry of Education established the off-campus education and training supervision department, which is mainly responsible for the specific implementation of this policy. After the implementation of the policy, the extracurricular tuition carried out by private training institutions have been significantly reduced, while the after-school services in charge of primary and secondary schools have replaced them, providing academic guidance and comprehensive quality and ability training for primary and secondary students.) on further easing the burden of excessive homework and off-campus tutoring for students undergoing compulsory education (“the Opinion”). The work of “double reduction” is conducted as follows: within the off-campus domain, bring order out of chaos by developing extracurricular tuition organizations in a proper, standardized way; within the on-campus domain, encourage schools to offer after-class service which offers targeted, diverse tutoring, and extended activities [3]. According to the Policy, necessary and proper after-class tutoring remains a beneficial supplement to students’ general learning process. The core, however, is to find out the key factors in the optimal starting process of after-class tutoring and their effect mechanism. “Brand new” high-quality after-class service should be built to overcome the previous deficiencies in extracurricular tuition. Therefore, from the perspective of policy evaluation, at the preliminary stage without abundant data and cases, we evaluated extracurricular tuition and its effect mechanism before the Policy was implemented and summarized the “shortboard” and the crux of extracurricular tuition in the past. This could be a reference for improving the after-class tutoring service and an important practice of raising the efficiency of the Policy and developing high-quality elementary education.

Concerning extracurricular tuition and its effect mechanism, the academia widely recognizes that extracurricular tuition affects academic performance. Nevertheless, many studies focus on whether the effect is affected by external factors. Additionally, scholars share different opinions and there is a lack of research with casual inference. As for questions such as, “Are there any differences between extracurricular tuition intensities?” and “Does and how does extracurricular tuition affect academic performance?”, such a research area remains a “black box” due to existing insufficient theories and empirical studies. Therefore, our study utilized CEPS2014-2015 to construct the framework of the ET-AP effect mechanism. With the controlled heterogeneity of junior middle school students, we discussed the causal correlation between extracurricular tuition and academic performance, and the intermediate effect of student individuals’ psychological factors. We hoped our study could offer a reference for the effective implementation of the Policy.

## 2. Literature Review and Hypotheses

### 2.1. Theoretical Base and Effect of Extracurricular Tuition

Researchers have conducted numerous studies on extracurricular tuition [4]. Stevenson and Baker found that the content and form of extracurricular tuition in a country are strongly correlated with the national distribution of education resources and curricular setting. Their relationship is similar to a man and his shadow. Therefore, Stevenson and Baker analogized extracurricular tuition to “shadow education [1]”. Before the implementation of the Policy, extracurricular tuition in the context of Chinese scholars mainly indicated extracurricular tuition in elementary education. Extracurricular tuition can also be regarded as a market-oriented economic behavior [5]. In China, gaokao (national college entrance exam, NCEE) enjoys a supreme status in the Chinese education system, and the saying “*yi kao ding zhongshen*” (one exam in a lifetime) prevails among the public [6]. For this reason, in the country where examinations are deemed the nucleus of education resource distribution [7], the notion “an early bird catches the worm” is deep-seated among parents. They would register tuition courses for their children in junior middle school or even primary school. Consequently, extracurricular tuition offered by off-campus organizations had boomed and flourished without inhibition [8] before the Policy was implemented.

A subsequent question is, what effect will this academic-race-oriented extracurricular tuition have on students’ academic performance? Some scholars found that the effect is positive. Lei and Hu et al., recognized such a positive effect and suggested that China invest more in extracurricular tuition [9,10]. Atalmis et al., conducted research on Turkish students and discovered that private tuition institutes improve students’ performance in mathematics [11]. Li’s research on students in the fourth grade in central China showed the same result: attending extracurricular tuition courses can improve students’ academic performance [12]. However, opponents held that, if imposed at every educational level, extracurricular tuition may have an effect that is below expectation. Roland and Euston studied the Singaporean tuition market. They found that excessive private tuition institutes, especially those offering one-on-one services, impede, rather than boost, academic performance [13]. Early in 2014, Song and Yang conducted a sampling survey on 180 middle and elementary schools of various types at the stage of compulsory education in 20 provinces (cities) within China. They concluded that more than half of the students in middle and elementary schools have a heavy academic burden [14]. In fact, the recent tuition-mania has proven to be an increased burden on many students. Tang and Fu debunked a widely-accepted opinion that increasing the length and amount of learning can improve academic performance. They saw a slight, if not none, impact of a heavier coursework burden on academic performance. Likewise, notwithstanding the positive correlation between learning duration and academic performance, excessive learning duration would be as the saying, “going beyond the limit is as bad as falling short” [15]. As a non-official paid education form or activity aiming to remedy shortcomings or to act as an excellence cultivation, extracurricular tuition distinguishes from and supplements mainstream school education [16]. Xue and Fang believe that fierce competition in extracurricular tuition among students and parents not only aggravates academic and economic burdens, but also interferes with the healthy development of school education. Such a competition may affect school education [17]. Before the Policy, extracurricular tuition was defined as an addition to and an enhancement of school curricula. However, excessive extracurricular tuition may hinder students from taking official courses, which scarcely contributes to their academic performance.

To sum up, from the angle of value-added evaluation, ceteris paribus, extracurricular tuition within a duration improves academic performance. Extracurricular tuition can, in certain conditions, add value to a student’s current academic performance [18]. Therefore, it is believed that regardless of students’ current academic performance, there is a motive and need to improve it through tuition. Based on the marginal utility theory, the degree of positive effect and the possibility of a negative effect from extracurricular tuition depends on whether the extracurricular tuition intensity can fall into a reasonable zone. In other words, a moderate extracurricular tuition intensity has a positive correlation with academic performance. However, ceteris paribus, when the intensity is high, the academic performance increment incurred by each additional unit of extracurricular tuition will diminish [19]. In other words, the positive correlation between extracurricular tuition and academic performance will see a marginal decline. Hence, for the academic performance increment, there exists a “turning point” of extracurricular tuition intensity, namely the threshold value, where the intensity has a positive effect on academic performance. The “optimal starting zone” of extracurricular tuition is therefore constructed.

Our study made the following hypotheses:

**H1:** 
*Extracurricular tuition can improve academic performance; extracurricular tuition intensity is not significant to academic performance.*


**H1_a_:** 
*The academic performance of students with extracurricular tuition is significantly higher than that of students without extracurricular tuition.*


**H1_b_:** 
*The academic performance of students with high extracurricular tuition intensity has no significant difference from that of students with low extracurricular tuition intensity.*


### 2.2. ET-AP Effect Mechanism in the Perspective of Behavioral Public Management

Apart from the effect of extracurricular tuition intensity on academic performance, our study also probed into the ET-AP effect mechanism, in other words, through what channel does extracurricular tuition strengthen the kinds of student capabilities to improve academic performance? Through the literature review, we found that most existing studies on the extracurricular tuition mechanism discussed the way extracurricular tuition as an intermediate variable improves academic performance. In their research on whether extracurricular tuition helps high school students in gaokao, Xue and Zhao set a prerequisite that extracurricular tuition can improve academic performance [20]. In their study on parent involvement, extracurricular tuition, and academic performance of middle school students, Li and Xue viewed extracurricular tuition as an educational investment that is critical for academic performance [21]. Besides, some focused on the inherent factors between extracurricular tuition and academic performance. Fang proved that interest-based extracurricular tuition can significantly improve a student’s non-cognitive ability in expression, reaction, and learning [22]. The research provided a feasible pathway to interpret the ET-AP effect mechanism, yet it did not consider the inner- and outer-environmental interaction.

Our study reviewed the above question from the perspective of behavioral public management. From this perspective, the behavioral public policy (BPP) exploits a targeted and science-based research pathway to raising policy efficiency. Using the research paradigm of behavioral science, BPP is evidence-informed research thinking to interpret the efficiency of macro public policies from the behaviors of micro individuals (e.g., the public) [23]. It lays a specific, operatable, and science-based foundation for maximizing policy efficiency [24]. Based on this framework, our study took extracurricular tuition as a behavioral event in the inner- and outer-environmental interaction. We therefore held that psychological quality is a critical factor between extracurricular tuition and academic performance. When taking extracurricular tuition courses, students will confront some form of academic stress as extracurricular tuition enhances [25]. They may also have outer-environmental stress, since extracurricular tuition, as an external academic training, mirrors family anxiety in education and parental expectation [2,26]. Psychological resilience indicates the effectiveness of a student individual’s stress tackling in the field of learning in the inner-outer-environmental interaction. It is a crucial tool to raise the efficiency of extracurricular tuition. The role of psychological resilience in students’ learning process has attracted the attention of scholars for many years. According to Xi et al., psychological resilience refers to the phenomenon that an individual who went through or is going through severe stress/adversity develops a body and mind which is not damaged, or contrarily, which is consolidated, by the said stress/adversity [27]. Hu carried out an empirical study on 311 middle school students, proving the great significance of psychological resilience for academic performance [28]. Similarly, Xie et al., from the perspective of resilience, demonstrated that students’ resilience and capabilities are a vital way to improve their life satisfaction [29]. Our study held that psychological resilience stabilizes academic performance. Good psychological health can improve students’ learning status [30]. Evidently, psychological resilience plays a pivotal role in learning.

Our study analyzed the ET-AP effect mechanism with psychological resilience as the entry point. With reference to Richardson’s research model: when an individual is faced with a life stress event, s/he will maintain a balanced body and mind using all available resources and reconstruct his/her psychological functioning. If the individual manages to tackle the imbalance, s/he can secure and improve his/her psychological resilience, and thus will reach a new balance. If s/he cannot appropriately reconstruct his/her psychological functioning, it means the reconstruction has failed, and thus his/her body and mind will have unrecoverable damages (see Figure 1) [31]. Borrowing Richardson’s model of psychological resilience, we established our theoretical basis.

Based on the above theory, along with the traits and nature of extracurricular tuition, our study invented the extracurricular tuition-psychological resilience-academic performance effect mechanism (the ET-PR-AP effect mechanism) (see Figure 2). The extra burden of extracurricular tuition will bring non-negligible academic stress to a student. When extracurricular tuition intensity emerges and increases, extracurricular tuition may become a life stress event to break the balance of his/her psychological status. Such stress pushes the student to improve his/her psychological resilience in the psychological reconstruction to create a new balance, thus enhancing his/her learning ability and academic performance. By contrast, a student without sound capabilities of psychological adjustment may find it hard to tackle this issue. This will damage his/her body and mind, and hence affect his/her academic performance. Zhang and Shen verified that the rise of academic stress may have a negative impact on a student’s academic performance [25]. On this account, our study explored the “black box” of the research on education effect mechanism in the psychological aspect by setting psychological resilience as an intermediate variable in the ET-AP effect mechanism. We raised the following hypothesis:

**H2:** 
*Psychological resilience has an intermediate effect in the ET-AP effect mechanism.*


To explore the ET-AP effect mechanism, we categorized extracurricular tuition intensity into high and low. We then discussed their effects on academic performance and the intermediate effect of psychological resilience on the two categories separately. Finally, we built the ET-AP effect mechanism from the perspective of behavioral public policy (see Figure 3).

## 3. Research Design

### 3.1. Data Source and Research Variables

To investigate the ETI-AP effect heterogeneity and the effect of psychological resilience in this pathway, our study borrowed and analyzed data on school administrators, teachers, students, and parents from 112 junior high schools in China from CEPS2014-2015. Considering the data in the second-wave longitudinal study, namely the data on students in the eighth grade (formerly seventh grade) which were optimized compared with the previous year, we screened out and analyzed them to keep sample consistency.

To reflect the effect of extracurricular tuition (ET) on the dependent variable, we defined academic performance (AP) as a student’s overall score (OS) in the name of a proxy variable—his/her standardized overall score of the Chinese, Math, and English disciplines in 2014. This step aimed to explore the ETI-AP effect and the disciplinary heterogeneity of the effect pathway.

For the independent variables, we selected the CEPS question “Does a student participate in extracurricular tuition courses?” as the indicator of ET. If a student participates in extracurricular tuition courses, then ET is 1, otherwise it is 0. To study whether the time and intensity of extracurricular tuition will influence the dependent variable, we established two variables: extracurricular tuition time (ETT) and extracurricular tuition intensity (ETI). ETT was established according to a student’s weekly extracurricular tuition time. ETI, a categorical variable, was established as follows. In need of categorization, we set an equation: ETI = ETT/20. According to the distribution of data on ETT of the CEPS samples in descriptive statistics, we set ETI = 0.2 (namely ETT = 4) as the high-low critical point. We defined ETI > 0.2 as high, assigning the value 2, and 0.2 ≥ ETI > 0 as low, assigning the value 1.

As for the intermediate variable, by referring to previous research [32], we included the psychological factors of a student individual in our research to draw a more reliable conclusion. In detail, we referred to the scales of will power (C24) and of psychological adjustment (C25) in the CEPS as indexes of psychological resilience (PR). We then quantified a student’s psychological resilience from will power (“Even though not feeling good, or there are other excuses to stay at home, I will still go to school”, “Even though I dislike the homework, I will still finish it as much as I can”, “Even though it takes a long time to finish the homework, I will continue to finish it as much as I can”, “I can continue to develop my interests and hobbies”) and psychological adjustment (“When having negative feeling, I can quickly adjust myself to it”, “When having negative feeling, I can quickly adjust myself to it with others’ help”.).

Concerning control variables, by referring to previous research [33], we selected routine demographic variables in research on extracurricular tuition including “Hukou—local resident”, “Gender”, “Ethnic group”, “Only child”, “Health status”, “Economic conditions”, “Education capital”, “Nature of school”, and “Teaching quality” as control variables. Besides, given the special trait of extracurricular tuition, and according to the research on the relationship between students’ sleep quality and their academic performance conducted by Gruber et al., we also included sleep quality as a control variable [34]. Finally, previous research showed that cognitive competence is closely related to academic performance [35], and some even equated the two [36]. This indicated the pivotal role of cognitive competence in academic performance. Cognitive competence may be an important variable on academic performance, and thus should be controlled. We set the CEPS scales of will power and of psychological adjustment as two indexes of psychological resilience. The quizzes do not involve memory knowledge that is taught in school curricula. Instead, they test a student’s logic thinking and problem-solving ability from three aspects: language, figures, calculation, and logic. The scales are internationally comparable and nationally standardized [35]; therefore, the score can reveal a student’s cognitive competence and is distinct from the academic score. Finally, we added 11 control variables, which are “Cognitive competence”, “Hukou—local resident”, “Gender”, “Ethnic group”, “Only child”, “Health status”, “Sleep quality”, “Economic conditions”, “Education capital”, “Nature of school”, and “Teaching quality” from three aspects, namely student, family, and school, to the model. All variables have been listed as shown in Table 1 as below:

The specific descriptive statistical results are shown in Table 2. The total sample quantity is 8090, with genders evenly distributed. Most samples lived in rural areas, in average economic conditions and of good health status. The proportion of only children taking extracurricular tuition courses is high.

### 3.2. Research Process

Our data analysis focused on the ET-AP effect mechanism and the intermediate effect of psychological resilience therein. We adopted the fixed-effect (FE) model, the regression analysis, the propensity score matching (PSM) method, and the intermediate effect model in our study. The study consisted of the following three steps:

Firstly, we set academic performance as a dependent variable and extracurricular tuition as independent variables. We also included different control variables. Then, we controlled the fixed effects of school and conducted a regression analysis. Based on the regression results, we tested the ET-AP influence by virtue of PSM. We put “non-ET” in the control group and “ET” in the experimental group. Then, we explored the treatment effect of extracurricular tuition on academic performance while controlling other endogenous variables.

Next, we probed into the effect mechanism of different extracurricular tuition intensities on academic performance. To be specific, we put “non-ET” in the control group and “ET” in the experimental group; and put “Low-ETI” in the control group and “High-ETI” in the experimental group. From the two group pairs, we acquired different extracurricular tuition intensity-academic performance effects (ETI-AP effects). Then, we further segmented and compared the extracurricular tuition intensities to identify an “optimal starting zone (OSZ)”.

Lastly, we referred to the analysis in the research on the “carbon reduction” policy by Kang et al., which did not extend the endogeneity issues in the intermediate mechanism [37]. Having discussed such issues, we directly set “ET” and “ETT” as independent variables. Then, we explored the ETI-AP effect mechanism and tested the existence and robustness of the intermediate effect of psychological resilience therein.

## 4. Empirical Analysis

### 4.1. Verification of the Effect of Extracurricular Tuition

In the study on extracurricular tuition and academic performance, one can observe the internal mechanism with the traditional ordinary least squares regression (OLS) which was adopted in previous studies [38,39,40]. Nonetheless, the model only includes partial observable variables; thus, it can neither eliminate the influence of endogenous variables on the regression results nor control the fixed effects. Hence, the model is prone to estimation errors and cannot effectively verify the causal correlation between variables [41]. To tackle this issue, we erased the estimation errors induced by self-selection bias by combining FE and PSM, to establish an accurate, science-based model. PSM was first introduced by Rosenbaum and Rubin in biological statistics [42]. By setting a series of hypotheses, PSM can calculate and obtain unbiased estimation results of average treated effect (ATT). This enables non-random observative research to produce “clean”, unbiased results like a natural experiment does.

Table 3 presents the results of the effects of different factors on academic performance for different samples. For Model (1), extracurricular tuition has a positive correlation with academic performance at 1% significance level, showing that extracurricular tuition can significantly improve academic performance. In order to verify the effect of extracurricular tuition on academic performance from diverse angles, our study regressed the continuous variable ETT and categorical variable ETI, respectively, as are shown in Models (2)–(5). For Models (2) and (3), we limited the samples to extracurricular tuition students and regressed ETT and ETI, respectively. We found that none of them had a significance level (*p* > 0.1) of regression on academic performance. This illustrates that with extracurricular tuition provided, the rise of extracurricular tuition time (extracurricular tuition intensity) cannot further promote academic performance. For Models (4) and (5), we discussed the effects of extracurricular tuition time with low and high extracurricular tuition intensities on academic performance, respectively. The results show that extracurricular tuition time with either low or high extracurricular tuition intensity does not have a significance of regression on academic performance. Therefore, for samples with extracurricular tuition, there is not much difference between high and low extracurricular tuition intensities.

The bi-grouping test of low and high extracurricular tuition intensities may not comprehensively reveal the marginal changes of the extracurricular tuition effect. Therefore, with reference to previous studies [43], we further segmented the samples according to extracurricular tuition intensity. We placed non-ET samples in the control group, samples with 0.2 ≥ ETI > 0 in Group 1, samples with 0.3 ≥ ETI > 0.2 in Group 2, samples with 0.4 ≥ ETI > 0.3 in Group 3, and samples with ETI > 0.4 in Group 4. Then, we set Group 1 (0.2 ≥ ETI > 0) as the control group and compared the differences between Group 2, 3, 4, and Group 1 to identify the OSZ. The results are shown in Table 4.

For Model (6), the results show an insignificant group difference, indicating no significant difference in extracurricular tuition effect between Group 1 and 2. For Model (7), the difference between Group 1 and 3 is significantly positive, reaching a significance level of 0.01. This indicates that the extracurricular tuition effect with ETI = 0.3–0.4 is higher than that with ETI = 0–0.2. When the extracurricular tuition intensity falls in this zone, extracurricular tuition has a stronger effect on academic performance than that with low extracurricular tuition time. This accords with the process from quantitative to qualitative changes. A small rise of extracurricular tuition intensity hardly improves the extracurricular tuition effect. Only when extracurricular tuition intensity increases continuously beyond a certain scope can it significantly improve the extracurricular tuition effect. For Model (8), the regression coefficient is negative, and the regression does not reach a significance level of 0.10. This shows that the extracurricular tuition effect does not improve as extracurricular tuition time continues to increase. When ETI = 0.4, the extracurricular tuition effect is, on the contrary, not higher than the effect with low extracurricular tuition intensity. This indicates that beyond the OSZ, excessive extracurricular tuition time will backfire. Thus, the OSZ should be that in Group 3, with an ETI of 0.3–0.4.

### 4.2. PSM-Based Robustness Test

Our study discussed the influence of extracurricular tuition on academic performance. To test the robustness of the ET-AP effect mechanism, we adopted PSM to verify the above conclusions based on OLS. Firstly, to minimize the omitted variables of the regression model, we regarded factors affecting academic performance as control variables and included them in the logit regression equation to calculate the propensity score. The score is shown in Table 5. Most samples fall into the common value range, with only 17 out of 8030 as exceptions. This is in line with the common support hypothesis. Then, we conducted a balancing test on each control variable to ensure the heterogeneity of the matched results, as is shown in Figure 4. Figure 4 shows that the standardized bias of each matched result is below 20%—a satisfactory outcome. After matching, the T values of the experimental group and the control group are not significant. That is to say, the control variables in the two groups have no systematic difference. The propensity score of each group is shown in Figure 4.

To produce more reliable PSM results, we calculated the data with one-to-one matching, radius matching, kernel matching, and nearest neighbor matching. The ATT values calculated are shown in Table 6. Before matching, ATT = 17.12 and T = 13.34 (*p* < 0.01). With one-to-one matching, ATT = 5.36 and T = 3.05 (*p* < 0.01). With radius matching, ATT = 16.80 and T = 12.88 (*p* < 0.01). With kernel matching, ATT = 4.77 and T = 3.20 (*p* < 0.01). With nearest neighbor matching, ATT = 5.36 and T =2.75 (*p* < 0.01). Evidently, the matched ATT values are lower than that before matching and reach the significance level of 1%. To sum up, although the ET-AP effect before matching is overestimated, it is still significant. In other words, H1a is tenable. However, the ETI-AP correlation and its effect mechanism should be further explored.

Next, our study tested the ETI-AP effect. The above research indicated that extracurricular tuition has a significant effect on academic performance. However, as extracurricular tuition intensity increases, can the effect maintain its robustness, or will it follow the law of diminishing marginal utility? This question is still to be answered. Therefore, by simulating Models (6)–(8), we conducted the robustness test by comparing the effects of different extracurricular tuition intensities with PSM. The process is as follows. Firstly, to minimize the omitted variables of the regression model, we regarded factors affecting academic performance as control variables and included them in the logit regression equation to calculate the propensity score. Then, we conducted a balancing test on each control variable to ensure the heterogeneity of the matched results. The results before and after matching are consistent, namely the extracurricular tuition effects of Groups 1 and 2, and of Group 1 and 4, have no significant difference. The extracurricular tuition effects of Groups 1 and 3 have significantly positive difference, namely the OSZ falls into Group 3 (with an ETI of 0.3–0.4). The regression results based on OLS are robust. To sum up, the matched results demonstrate that extracurricular tuition does significantly improve academic performance. However, high and low extracurricular tuition intensities have similar effects on academic performance. That is to say, high extracurricular tuition intensity has no significance for academic performance and may even weaken or eliminate the extracurricular tuition effect on academic performance. H1b is tenable.

### 4.3. Mediation Effect of Psychological Resilience

To further explore the ET-AP effect mechanism, we built an intermediate model to identify the mechanism’s two indirect effects. To verify the effect of psychological resilience in the ET-PR-AP pathway, we set “ET” and “ETT” as independent variables, “PR” as intermediate variable, and “AP” as dependent variable. We also added a series of control variables to the model. In the aspect of extracurricular tuition, the results are shown in Table 7. The ET-AP overall effect is 6.448 (*p* < 0.01), the ET-PR-AP intermediate effect is 1.045 (a = 0.096, *p* < 0.01; b = 10.891, *p* < 0.01), the direct effect of extracurricular tuition is 5.402 (*p* < 0.01), and the proportion of intermediate effect of psychological resilience is 16.21%. This indicates that psychological resilience plays an intermediate role in the ET-AP effect mechanism. In other words, extracurricular tuition itself can improve academic performance by cultivating a student’s psychological resilience.

## 5. Discussion

Taking advantage of PSM, we studied the ET-AP effect mechanism of junior middle school students and the intermediate effect of psychological resilience therein. We found that: (1) extracurricular tuition does improve academic performance; (2) high and low extracurricular tuition intensities have a similar effect on academic performance, and there is an “optimal starting zone” of extracurricular tuition; and (3) considering the result (yes–no), or the degree (high–low), psychological resilience has different intermediate effects in the mechanism. Based on these findings, we concluded our discussion as follows.

### 5.1. The Learning Demand Reflected from Extracurricular Tuition Is an Objective, Non-Negligible Reality

Having controlled other influence factors, our study discovered that students who take extracurricular tuition courses do have better academic performance than those who do not. In other words, extracurricular tuition promotes academic performance. Theoretically, the implementation of the “double reduction” policy is a powerful way to alleviate coursework burden, hence reducing students’ academic stress. Practically, however, in the current education environment that is directed by zhongkao (senior high school entrance exam) and gaokao, extracurricular tuition is a spontaneous behavior of students and parents in the competition environment.

There are many reasons for the need of extracurricular tuition. First, the traditional school classroom education is oriented to class teaching, and it is difficult to carry out targeted management on the learning situation of each student. Compared with school classroom teaching, extracurricular tuition in the form of one-to-one or small-class teaching can more accurately find out the knowledge points that each student fails to grasp in classroom learning and provide targeted guidance to fill in the gaps. Secondly, in traditional Chinese culture, education has been endowed with the moral and political functions of cultivating personal accomplishment and educating the people: “jade is not polished, it is not a tool; people do not learn, he is ignorant. It is therefore the ancient king who establishes the country and the people, teaching is the first” (people will not understand the Confucian doctrine without learning. Therefore, the ancient king, to establish the country and rule the people, should take education as the primary task) (“Learning Record”). The Chinese Imperial Examination System closely links the level of education with personal career development and the rise of social class [44]. The modern gaokao national examination system has similar functions, which leads to the general demand for shadow education in Chinese society. Finally, on the demand side, the theory of effectively maintained inequality (EMI) considers that in order to maintain the class status, families of the superior class can only seek extracurricular supplementary education outside of the equalization of compulsory education, so as to improve the competitiveness of their children in school [45]. Therefore, this social value has undeniable significance for academic development. The strong market of extracurricular tuition cannot be simply inhibited by policies. In essence, the demand of students and parents implies students’ desire for in-depth learning of textbook knowledge to enhance their competitiveness in their future competition.

Nevertheless, this objective demand has become the rooted obstacle against the implementation of the Policy to some extent. According to the theories of the behavioral public policy, to successfully implement the Policy, policymakers should put themselves in the shoes of students and parents. Only when students and parents understand and accept the care for students’ physical and psychological health behind the policy will they obey it. Currently, the demand for further study is the barricade against people’s recognition of the Policy. In our opinion, albeit the proposal of regulating discipline-based extracurricular tuition, policymakers should also recognize the positive effect of extracurricular tuition and transplant it into school education and after-class service. The multiple channels built therefrom will satisfy students and parents’ demands and boost school education. Such is a necessity to implement the “double reduction” policy.

### 5.2. There Is an “Optimal Starting Zone” in the Extracurricular Tuition Effect

We compared high and low extracurricular tuition intensities, as well as cases with and without extracurricular tuition. It is known that extracurricular tuition can improve academic performance. Nonetheless, such is not a purely linear relationship—there exists an “optimal starting zone”. In other words, to make extracurricular tuition effective, one should respect the law of the transformation of quantity into quality. More importantly, s/he should pay attention to excessive extracurricular tuition [46] and maintain the positive effect of extracurricular tuition at a critical point. Beyond the critical point, the positive correlation between extracurricular tuition intensity and academic performance will suspend. Increasing the intensity will not improve the effect of extracurricular tuition. This study finds that the effect reaches the best state when the extracurricular tuition time is up to 8 h per week. If the time continues to be extended, the improvement effect will gradually weaken and ultimately disappear. In other words, high extracurricular tuition intensity does no good to academic performance. There is a marginal effect of extracurricular tuition on academic performance.

We believe the above discussion will enlighten the policy implementation. Although extracurricular tuition has switched from off-campus to on-campus, in nature, it is still a supplementary form of after-class learning, with an “optimal starting zone” to improve academic performance. Students should maintain a certain learning intensity off the campus to improve their academic performance. However, the intensity should fall into an optimal zone to take effect. Compared with senior middle schools, junior middle schools emphasize more on cultivating students’ learning habits and attitudes [47]. Low extracurricular tuition intensity can hardly guarantee the quantitative–qualitative transition. By contrast, high intensity may occupy students with coursework burden, triggering their psychological inversion and academic burnout: it will consume their good learning attitudes and thus reduce the effect of extracurricular tuition. Moreover, the most effective status of each individual may be different due to the unique personality characteristics, physical and mental development, learning basis, and other factors. The central government provides macro guidance for all regions to promote the rationalization and standardized development of extracurricular tuition. Under the macro guidance of the “double reduction policy”, all regions have accumulated some experience in the reasonable regulation of extracurricular activities, and constantly explore the after-school education mode. The best effective mechanism obtained in this study can provide a certain reference to determine the reasonable range of extracurricular tuition time. Therefore, extracurricular tuition does not take effect by blindly increasing time and intensity. It is necessary to explore alternatives to enhance quality and quantity from multiple channels, since a suitable atmosphere for extracurricular tuition will bring more benefits to students’ academic performance.

### 5.3. Psychological Resilience Is the “Conversion Valve” of Extracurricular Tuition

Since the implementation of the fourteenth five-year plan, educational development with quality has become an urgent core task in China’s elementary education. To achieve this goal, it is necessary to understand the education mechanism. According to our ET-PR-AP effect mechanism, extracurricular tuition is an education form with the nature of academic stress event, whereas psychological resilience is the “conversion valve” of the effect of extracurricular tuition. From the perspective of behavioral public management, psychological resilience is an indicator of the effectiveness of an individual student’s stress tackling in the inner- and outer-environmental interaction. It determines whether the “double reduction” policy can benefit students and parents.

In the ET-PR-AP effect mechanism, extracurricular tuition as an additional learning event is an academic stress event from the inner- and outer-environments for students, which has the possibility to cause dysfunction. In this context, psychological functioning may be reconstructed in three directions, namely, consolidating, maintaining, or damaging the psychological resilience. The reconstruction direction depends on an individual’s psychological resilience and the level of the stress event. With psychological resilience unchanged, low extracurricular tuition intensity will cultivate students’ psychological resilience and thus improve their academic performance. High intensity, however, will impose a heavier academic burden, hurting the psychological resilience of students who have not developed sound psychological adjustment. This may damage students’ body and mind and do little good to their academic performance. In an interview, a parent of a junior middle school student expressed similar views, acknowledging the promotion effect of extracurricular tuition while worrying about the psychological stress brought by high extracurricular tuition intensity (20200924Z). Much evidence proves that whatever the stage of extracurricular tuition, psychological resilience is the prerequisite to improve academic performance with extracurricular tuition. Cultivating students’ psychological resilience, therefore, is a feasible way to develop China’s elementary education with quality.

## 6. Policy Implications

According to the above data analysis and discussion results, we raised three suggestions on the “double reduction” policy against extracurricular tuition.

Firstly, replace extracurricular tuition with after-class service to improve students’ learning quality. Since the Policy was implemented, most primary and junior middle schools in China have begun to offer “5 + 2 (five days a week, two hours a day)” after-class tutoring service. Nevertheless, the arrangement for schoolteachers and curricula have not been standardized, which results in deficiencies in after-class service. For example, shift teachers find it hard to tutor students; students lack sufficient materials to fulfill their two-hour learning demand. Thus, in the forthcoming reform, it is suggested that such problems be solved to ameliorate the system of after-class service as an alternative to extracurricular tuition.

In terms of teaching technology, this study helps to improve the after-school service system of the school, introduce the teaching method represented by Universal Design for Learning, develop a flexible teaching environment, accept teaching feedback to solve the differences of students’ individual learning conditions, and help all students achieve efficient learning [48] so as to improve the after-school service quality and better perform the alternative role of after-school service for extracurricular remedial learning. In terms of the content of after-school activities, some schools hold campus poetry contests and other forms popular with students to combine the formal teaching content with excellent traditional culture education and consolidate classroom knowledge with after-school activities, which can not only improve the quality of after-school services, but also avoid blindly extending the learning time to bring heavy burden to students.

Secondly, pay attention to the negative effect of high extracurricular tuition intensity on students’ physical and psychological health as well as their academic performance and explore the “optimal starting zone” of extracurricular tuition in teaching practice. At present, discipline-based extracurricular tuition at the stage of elementary education is increasing, which lays a heavy burden on families and students. This is one of the reasons why the central government strengthens the regulations on extracurricular tuition. Our study proved that there exists an “optimal starting zone”, beyond which extracurricular tuition may be less effective on academic performance. Academic performance may not be necessarily better as extracurricular tuition increases. Nonetheless, from the investigation within Guangdong province, we knew that in some regions, junior high schools vie with each other. They join academic races, secretly offering tuition courses to students with no intention to stop. Such a prisoner’s dilemma hinders schools from following the “double reduction” policy. Therefore, schools themselves should seek to terminate such vicious contests. They should explore the “optimal starting zone” of the time of after-class service and extracurricular tuition that fits them best in the teaching practice; replace the increase of tuition with quality teaching, raise parents’ awareness of the objective restriction of tuition as well as the importance of moderate tuition and learning. In addition, the government should standardize the ecology of the education and training industry, regulate the length of extracurricular tuition, guide after-school tuition institutions to transform into non-disciplinary training institutions, and encourage the provision of quality education services that promote the comprehensive development of education.

Thirdly, strengthen the multi-party cultivation of psychological resilience and provide psychological support. The issuing of the Opinion implies that the improvement of education quality has become an urgent issue in China’s elementary education. Our study indicated that education quality can be improved by not only deepening and improving disciplinary knowledge, but also promoting students’ psychological development. Psychological resilience is an indispensable factor for students to make academic progress from extracurricular tuition. It is also crucial for their academic and professional development. While restricting extracurricular tuition, the government should also encourage family education and peer company. It is suggested to enrich the education environment through multi-party interaction and communication. For example, at present, some schools have integrated mental health education into after-school services and carried out team psychological counseling activities. This practice provides psychological counseling and help during after-school services so as to reduce “negative” feelings and “pressure” among students. This will broaden the channels for students to access psychological resources, improve their psychological resilience, and promote their psychological development, all of which helps cultivate all-round talents.

## Figures and Tables

**Figure 1 behavsci-13-00217-f001:**
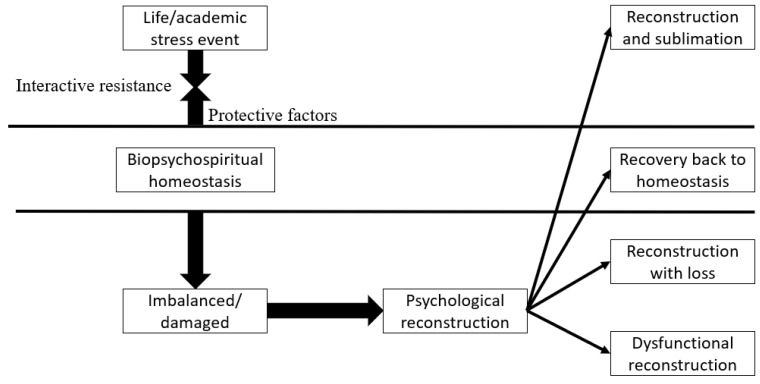
Model of psychological resilience [31].

**Figure 2 behavsci-13-00217-f002:**
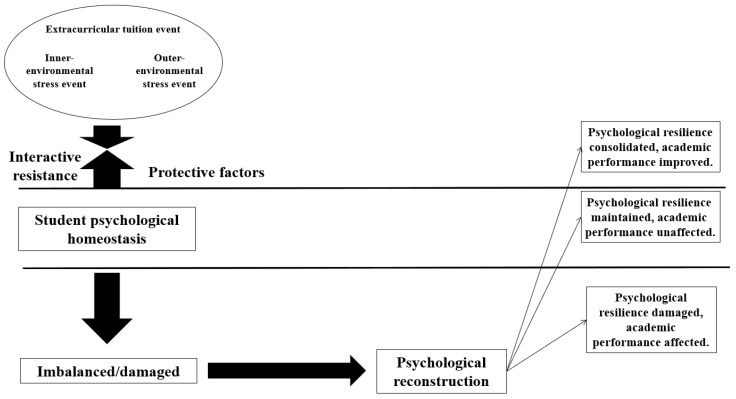
ET-PR-AP effect mechanism.

**Figure 3 behavsci-13-00217-f003:**
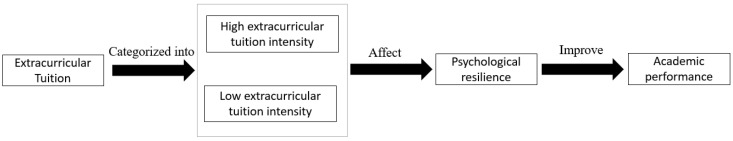
ET-AP effect mechanism.

**Figure 4 behavsci-13-00217-f004:**
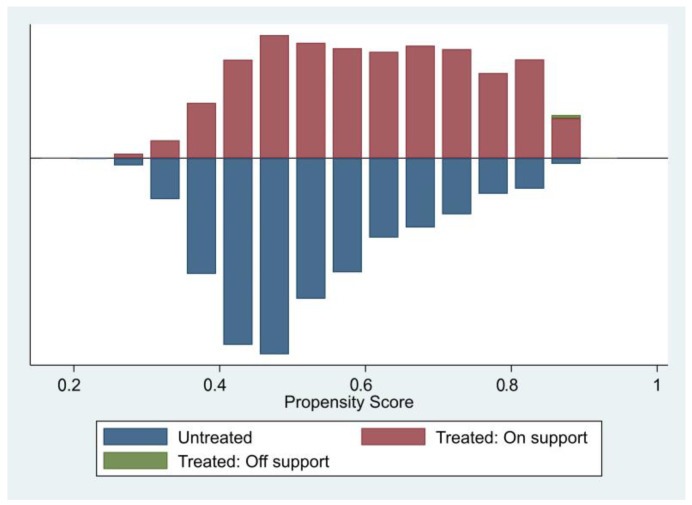
Common value range of propensity score (on and off support).

**Table 1 behavsci-13-00217-t001:** Definition and description of variables.

Type	Name	Definition and Description
Dependent variable(s)	AP—OS	Standardized overall score of the Chinese, Math, and English disciplines.
Independent variable(s)	ET	Does a student participate in extracurricular tuition courses? (0 = no; 1 = yes)
	ETT	Student weekly extracurricular tuition time.
	ETI	Student weekly extracurricular tuition intensity. (1: 0.2 ≥ ETI > 0; 2: ETI > 0.2)
	Sample treats (treat1, treat2, treat3, treat4)	Does the sample’s extracurricular tuition intensity fall into a certain zone? (<0.2, 0.2–0.3, 0.3–0.4, >0.4) (0 = no; 1 = yes)
Intermediate variable(s)	PR	Indexes of student will power and psychological adjustment.
Control variable(s)	Student	
	Cognitive competence	Standardized score of the quizzes of cognitive competence.
	Hukou—local resident	Is a student a local resident? (0 = no; 1 = yes)
	Gender	Student gender. (0 = female; 1 = male)
	Ethnic group	Student ethnic group. (0 = minorities; 1 = Han)
	Only child	Is a student the only child? (0 = no; 1 = yes)
	Health status	Does a student often fall ill? (0 = bad; 1 = good)
	Sleep quality	Does a student have sleep problems? (0 = no; 1 = yes)
	Family	
	Economic conditions	Student family economic conditions. (1 = below average; 2 = average; 3 = above average)
	Education capital	Parents’ average education level (1 = low; 2 = average; 3 = high)
	School	
	Nature of school	Is it a public school? (0 = non-public; 1 = public)
	Teaching quality	The school’s local ranking. (1 = lowest; 2 = below average; 3 = average; 4 = above average; 5 = highest)

Data source: Our originals.

**Table 2 behavsci-13-00217-t002:** Variable descriptive statistics.

Name	Non-ET (N = 3423)				ET (N = 4607)							
					Low-ETI (N = 2609)				High-ETI (N = 1998)			
	M	SD	Min	Max	M	SD	Min	Max	M	SD	Min	Max
ETT					2.46	1.06	1	4	7.41	2.67	5	20
AP	186.54	60.28	5.00	290.00	201.81	56.03	0.00	297.00	206.07	51.55	22.00	293.33
PR	2.97	0.60	1.00	4.00	3.09	0.56	1.00	4.00	3.12	0.59	1.00	4.00
Cognitive competence	0.25	0.82	−3.14	2.06	0.38	0.80	−2.70	2.04	0.41	0.78	−2.75	2.06
Gender	0.49	0.50	0.00	1.00	0.45	0.50	0.00	1.00	0.45	0.50	0.00	1.00
Hukou—local resident	0.16	0.37	0.00	1.00	0.20	0.40	0.00	1.00	0.27	0.45	0.00	1.00
Ethnic group	0.92	0.28	0.00	1.00	0.93	0.26	0.00	1.00	0.94	0.24	0.00	1.00
Only child	0.35	0.48	0.00	1.00	0.47	0.50	0.00	1.00	0.60	0.49	0.00	1.00
Health status	0.91	0.29	0.00	1.00	0.92	0.26	0.00	1.00	0.90	0.30	0.00	1.00
Economic conditions	1.78	0.52	1.00	3.00	1.87	0.50	1.00	3.00	1.97	0.47	1.00	3.00
Education capital	1.38	0.59	1.00	3.00	1.63	0.73	1.00	3.00	1.89	0.77	1.00	3.00
Nature of school	0.92	0.27	0.00	1.00	0.94	0.25	0.00	1.00	0.96	0.20	0.00	1.00
Teaching quality	3.88	0.86	1.00	5.00	4.03	0.85	1.00	5.00	4.16	0.78	1.00	5.00
Sleep quality	0.56	0.50	0.00	1.00	0.57	0.50	0.00	1.00	0.55	0.50	0.00	1.00

**Table 3 behavsci-13-00217-t003:** ET-AP regression results.

	(1)	(2)	(3)	(4)	(5)
	AP (Full Sample)	AP (ET)	AP (ET)	AP (Low-ETI)	AP (High-ETI)
ET (control group: non-ET)	4.293 ***				
	(0.890)				
ETT		−1.301		14.981	−9.326
		(3.513)		(13.674)	(5.997)
High-ETI (control group: Low-ETI)			0.679		
			(1.097)		
PR	10.171 ***	9.079 ***	9.059 ***	9.467 ***	8.570 ***
	(0.724)	(0.940)	(0.939)	(1.307)	(1.389)
Cognitive Competence	37.102 ***	34.684 ***	34.750 ***	35.864 ***	33.231 ***
	(0.594)	(0.780)	(0.780)	(1.056)	(1.186)
Gender	−18.400 ***	−16.509 ***	−16.531 ***	−18.269 ***	−14.295 ***
	(0.828)	(1.053)	(1.052)	(1.420)	(1.621)
Hukou—local resident	−3.006 ***	−2.435 *	−2.488 *	0.990	−5.727 ***
	(1.142)	(1.404)	(1.403)	(1.954)	(2.057)
Ethnic group	0.911	−2.677	−2.670	−7.358 *	3.245
	(2.210)	(2.797)	(2.797)	(3.971)	(4.005)
Only children	1.701 *	2.494 *	2.497 *	0.300	4.500 **
	(1.003)	(1.291)	(1.291)	(1.763)	(1.936)
Health status	−0.732	−0.759	−0.670	−0.800	−0.489
	(1.467)	(1.894)	(1.894)	(2.677)	(2.742)
Economic conditions	−1.114	−2.620 **	−2.649 **	0.327	−6.658 ***
	(0.882)	(1.154)	(1.155)	(1.537)	(1.801)
Education capital	4.035 ***	3.696 ***	3.661 ***	3.262 ***	4.016 ***
	(0.735)	(0.883)	(0.882)	(1.223)	(1.306)
Sleep quality	−0.957	−0.300	−0.258	0.207	−0.709
	(0.843)	(1.081)	(1.080)	(1.462)	(1.645)
_cons	155.020 ***	170.739 ***	170.209 ***	166.522 ***	177.699 ***
	(3.674)	(4.807)	(4.782)	(6.826)	(7.301)
School fixed effects	Controlled	Controlled	Controlled	Controlled	Controlled
Observations	8030	4607	4607	2609	1998
R-squared	0.401	0.368	0.368	0.387	0.351

Note: *** *p* < 0.01, ** *p* < 0.05, * *p* < 0.1.

**Table 4 behavsci-13-00217-t004:** Comparison of extracurricular tuition effects with different ETIs.

	(6)	(7)	(8)
	AP (Group 1 vs. 2)	AP (Group 1 vs. 3)	AP (Group 1 vs. 4)
Group difference	0.329	4.532 ***	−2.554
	(1.356)	(1.701)	(1.856)
_cons	167.134 ***	167.738 ***	171.862 ***
	(5.495)	(5.905)	(6.000)
Control variable	Controlled	Controlled	Controlled
School fixed effects	Controlled	Controlled	Controlled
Observations	3569	3159	3097
R-squared	0.373	0.383	0.380

Note: *** *p* < 0.01.

**Table 5 behavsci-13-00217-t005:** Sample balancing test (on and off support).

Variable	Experimental Group	Control Group	Standardized Bias %	T Value
Psychological resilience	3.10	3.10	−0.4	−0.19
Cognitive competence	0.39	0.36	3.5	1.69
Gender	0.45	0.43	3.1	1.47
Hukou—local resident	0.23	0.23	−1.8	−0.82
Ethnic group	0.93	0.93	−0.3	−0.17
Only children	0.53	0.51	2.9	1.34
Health status	0.92	0.91	0.6	0.30
Economic conditions	1.91	1.91	0.6	0.27
Education capital	1.74	1.75	−1.1	−0.48
Nature of school	0.94	0.93	4.4	2.18
Teaching quality	4.08	4.06	3.1	1.56
Sleep quality	0.56	0.55	2.4	1.13

**Table 6 behavsci-13-00217-t006:** ATT of extracurricular tuition in different matching methods.

	Sample Matching	Experimental Group	Control Group	ATT	Standardized Errors	T Value
Academic performance	Unmatched	203.66	186.54	17.12	1.28	13.34 ***
	One-to-one matching	203.57	198.21	5.36	1.76	3.05 ***
	Radius matching	203.57	186.76	16.80	1.30	12.88 ***
	Kernel matching	203.57	198.79	4.77	1.49	3.20 ***
	Nearest neighbor matching	203.57	198.21	5.36	1.95	2.75 ***

Note: *** *p* < 0.01.

**Table 7 behavsci-13-00217-t007:** Psychological resilience mediation effect (ET and non-ET).

	Full Samples		
	(12)	(13)	(14)
	AP	PR	AP
ET	6.448 ***	0.096 ***	5.402 ***
	(1.021)	(0.014)	(1.015)
PR			10.891 ***
			(0.866)
Control variable	Controlled	Controlled	Controlled
	131.142 ***	2.604 ***	102.786 ***
	(4.036)	(0.054)	(4.55)
Sample quantity	8030	8030	8030
R^2^	0.451	0.066	0.462

Note: *** *p* < 0.01.

## Data Availability

The data for this study can be downloaded from the official website of the China Education Panel Survey (CEPS): http://ceps.ruc.edu.cn/index.htm.

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
