# Peer review of "Effect Mechanism of Extracurricular Tuition and Implications on “Double Reduction” Policy: Extracurricular Tuition Intensity, Psychological Resilience, and Academic Performance"

_behavsci, 2023, doi:10.3390/bs13030217_

Round 1

Reviewer 1 Report

This appears to be a well written paper about a thoughtfully done study that appears to be relevant to certain educational contexts in specific countries.  Unfortunately, it was difficult for me to understand some of the paper because I live and work in a different context.  Even the term extracurricular tuition is not used in my context.  Each of the two terms are used, of course, but I could only guess at what they meant when combined into a single term.  My lack of cultural understanding also became a problem when you were describing some of the methodology of the study, e.g., when you talked about the data that were analyzed.  Consider using footnotes to explain such things to those from a different cultural context.  Other than that, the paper seemed to be competently done.  I just wish I could have better understood what you were writing about.  Despite excellent English construction, I often could not do this because of my lack of cultural knowledge.  

The design fit the purpose of the study and even included the propensity scoring technique which is far superior to the traditional matching technique when random assignment to control and experimental groups is not feasible (which is most of the time when research is conducted outside the laboratory). I also commented on the quality of the writing. It was clear and grammatically correct, but, of course. to be understandable, a reader must understand the cultural references in the writing or at least have such understanding provided, at the very least in footnotes. I simply could not comment on the applications/implications discussion because I did not have enough information about the cultural context in which the study's results could be applied.

Reviewer 2 Report

The topic is relevant and the manuscript is interesting.

The authors propose to make a discussion / evaluation of public policies. This aspect is referred to throughout the work, without, however, this discussion / evaluation is actually materialized. Rather, it justifies a given option.

It is important to clarify this aspect.

Reviewer 3 Report

I thoroughly enjoyed reading your manuscript. You were meticulous in the exploration of the topic, identification of all the potential variables, conducted a rigorous evaluation, and shared reasonable findings based on the data. Your study findings have utility not only in China, but other countries where extracurricular tutoring occurs. As you stated, it is the "shadow education" that is not regulated nor often evaluated. 

Your finding that high and low extracurricular tutoring were nearly similar was a surprise. Your finding that the middle way of moderate tutoring was most effective for the complex reasons that your research model evaluated made sense. My question is, how much is enough? How could you speculate on what the moderate approach looks like would be helpful? The other question is why your final set of policy recommendations did not raise the issue of why the primary instruction received by the students was insufficient and additional tutoring was needed. In other countries, pedagogies such as Universal Design for Learning are leading a revolution in how learning takes place in the classroom. Is this an issue that you could raise? Once again, an outstanding contribution to "edu-ecology" as you stated in your paper. Well done.
